# Mechanisms of Resistance and Implications for Treatment Strategies in Chronic Myeloid Leukaemia

**DOI:** 10.3390/cancers14143300

**Published:** 2022-07-06

**Authors:** Govinda Poudel, Molly G. Tolland, Timothy P. Hughes, Ilaria S. Pagani

**Affiliations:** 1Cancer Program, Precision Medicine Theme, South Australian Health and Medical Research Institute (SAHMRI), Adelaide, SA 5000, Australia; govinda.poudel@sahmri.com (G.P.); molly.tolland@sahmri.com (M.G.T.); tim.hughes@sahmri.com (T.P.H.); 2School of Medicine, Faculty of Health and Medical Sciences, University of Adelaide, Adelaide, SA 5000, Australia; 3Australasian Leukaemia and Lymphoma Group, Richmond, VIC 3121, Australia; 4Department of Haematology and Bone Marrow Transplantation, Royal Adelaide Hospital and SA Pathology, Adelaide, SA 5000, Australia

**Keywords:** chronic myeloid leukaemia, tyrosine kinase inhibitors, therapy resistance, BCR::ABL1-independent mechanism of TKI resistance, combination treatments

## Abstract

**Simple Summary:**

Chronic myeloid leukaemia (CML) is a type of blood cancer that is currently well-managed with drugs that target cancer-causing proteins. However, a significant proportion of CML patients do not respond to those drug treatments or relapse when they stop those drugs because the cancer cells in those patients stop relying on that protein and instead develop a new way to survive. Therefore, new treatment strategies may be necessary for those patients. In this review, we discuss those additional survival pathways and outline combination treatment strategies to increase responses and clinical outcomes, improving the lives of CML patients.

**Abstract:**

Tyrosine kinase inhibitors (TKIs) have revolutionised the management of chronic myeloid leukaemia (CML), with the disease now having a five-year survival rate over 80%. The primary focus in the treatment of CML has been on improving the specificity and potency of TKIs to inhibit the activation of the BCR::ABL1 kinase and/or overcoming resistance driven by mutations in the BCR::ABL1 oncogene. However, this approach may be limited in a significant proportion of patients who develop TKI resistance despite the effective inhibition of BCR::ABL1. These patients may require novel therapeutic strategies that target both BCR::ABL1-dependent and BCR::ABL1-independent mechanisms of resistance. The combination treatment strategies that target alternative survival signalling, which may contribute towards BCR::ABL1-independent resistance, could be a successful strategy for eradicating residual leukaemic cells and consequently increasing the response rate in CML patients.

## 1. Introduction

Chronic myeloid leukaemia (CML) is a malignancy characterized by the clonal proliferation of white blood cells that are mostly originated from the myeloid lineage in the bone marrow [1]. CML arises from the t(9;22)(q34;q11) balanced reciprocal translocation between chromosome 9 and 22 that forms the Philadelphia chromosome [2,3]. The translocation event results in the fusion of the *Breakpoint Cluster Region* (*BCR*) gene with the *Abelson proto-oncogene 1* (*ABL1*) gene, generating the *BCR::ABL1* fusion gene [4]. The resulting chimeric protein, BCR::ABL1, is a potent tyrosine-kinase signalling protein that drives cell proliferation and reduces apoptosis, which causes leukaemia [4]. Depending on the position of the *BCR* breakpoint, different BCR::ABL1 protein isoforms are generated (Figure 1A) [5]. The e13a2/e14a2 alternative transcripts (or b2a2/b3a2), resulting from the juxtaposition of *BCR* exon 13 or 14 with *ABL1* exon 2, produce a 210 kDa protein, which is found in over 90% of CML patients. The e1a2 transcript encodes a 190 kDa protein (p190), which is rare in CML but is relatively common in acute lymphoblastic leukaemia, occurring in around 70% of cases (Figure 1B) [5,6]. Approximately 95% of CML patients are diagnosed at the chronic phase, which is relatively indolent but can involve symptoms such as fatigue, abdominal pain, or weight loss. During this phase, the disease can be effectively managed with tyrosine kinase inhibitors (TKIs). If untreated, CML could progress to an accelerated phase (AP), which can last for up to a year, and eventually progress into the terminal blast phase of the disease, termed the blast crisis (BC). The blast crisis is characterized by the presence of excess blast cells in the blood or bone marrow [7]. Blast crisis results in dismal treatment outcomes, and it is often fatal even with intervention [7]. Patients in AP and BC are generally grouped together as advanced-phase CML patients [8].

The *BCR::ABL1* fusion gene is translated into the BCR::ABL1 protein, which contains several domains from BCR and ABL1. The BCR region of this protein regulates its enzymatic activity and provides sites for its binding partners [9,10]. The coil–coil domain of the BCR N-terminal part is responsible for the oligomerization and constitutive activation of BCR::ABL1 activity (Figure 2C) [11]. The ABL1 component of the BCR::ABL1 protein contains an SRC-homology-2 (SH2) domain, an SH3 domain, and a kinase domain [4]. When not fused with BCR, ABL1 has a myristoylated N-terminal responsible for the auto-inactivation of ABL1 kinase activity [4]. However, the myristoylated N-terminal is lost during the BCR::ABL1 fusion process [12]. The BCR:ABL1 kinase domain consists of key motifs responsible for its activity, including the phosphate binding loop (p-loop), the contact site (ATP/IM binding site), the catalytic domain, and activation loop (A-loop) (Figure 2C) [13]. BCR::ABL1 is active when ATP binds to the active site in the ABL1 kinase domain and transfers its phosphate group to ABL1 substrate. However, tyrosine kinase inhibitors (TKIs) compete with ATP for binding to the active site, inhibiting the BCR::ABL1 activation and preventing leukaemia (Figure 2A) [14].

The treatment of CML with TKIs has been paradigm-shifting, increasing the survival from 20% to over 80% today [14]. There are five available approved TKIs, prescribed based on disease phase, individual risk assessment, response level, presence of BCR::ABL1 kinase domain mutations, and response to prior TKI therapy [15,16]. The current available TKIs include the first-generation TKI imatinib (Glivec, Novartis, Basel, Switzerland ); second-generation TKIs dasatinib (Sprycel, Bristol-Myers Squibb, New York, USA), nilotinib (Tasigna, Novartis), and bosutinib (Bosulif, Pfizer, New York, USA); and third-generation ponatinib (Iclusig, Takeda/Incyte, Tokyo, Japan). Ponatinib is approved as a third-line treatment, when two or more TKIs are not effective, and for patients harbouring the T315I mutation in the BCR::ABL1 kinase domain. In October 2021, the Food and Drug Administration approved the new allosteric inhibitor of BCR::ABL1, asciminib (Scemblix, Novartis), for CML patients previously treated with two or more TKIs and for T315I mutations [17]. Asciminib binds to the myristoyl’s site on BCR::ABL1 and allosterically inhibits its activation, including that of BCR::ABL1 with T315I mutation, with very high selectivity, preventing downstream signalling [18]. ABL1 can normally self-regulate its activity via its engagement with the myristoylated N-terminal; however, in patients with CML, this ability is lost when ABL1 is fused with BCR (Figure 2B) [12]. Therefore, the binding of asciminib to a myristoyl pocket facilitates the inhibition of BCR::ABL1 activity by restoring its allosteric inhibition ability (Figure 2B).

**Figure 2 cancers-14-03300-f002:**
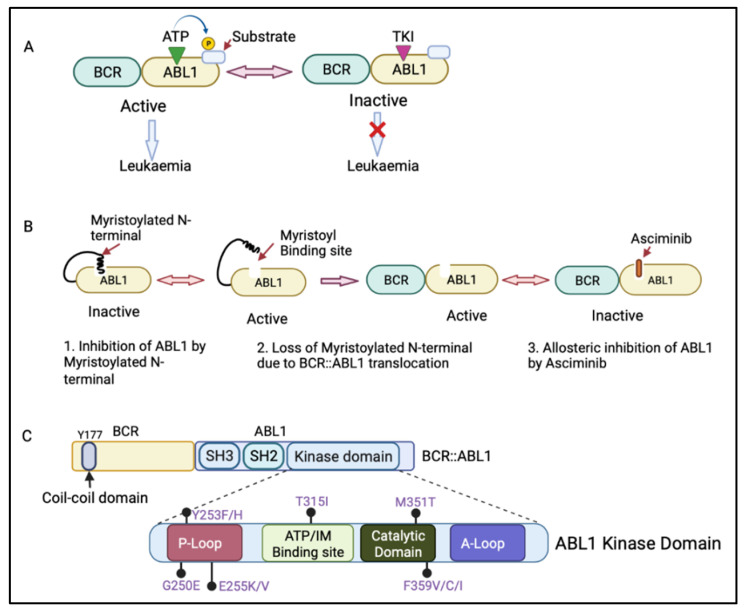
(**A**) Mechanism of action of adenosine triphosphate (ATP)-competitive tyrosine kinase inhibitors (TKIs). ATP binds to the ABL1 kinase domain, and the phosphate group is transferred to the ABL1 substrate, leading to BCR::ABL1 activation. The TKI competes with ATP for binding to the ABL1 kinase domain, inhibiting BCR::ABL1 activation and therefore preventing leukaemia progression. (**B**) Mechanism of action of asciminib, an allosteric inhibitor of BCR::ABL1. Asciminib binds to the myristoyl binding site, leading to BCR::ABL1 inactivation via allosteric inhibition of ABL1 kinase. (**C**) Schematic diagram of the BCR and ABL1 components of the BCR::ABL1 protein showing the N-terminal coil–coil domain (containing key tyrosine residue at 177 position, Y177) of BCR and an SRC-homology-2 (SH2) domain, an SH3 domain, and a kinase domain of ABL1. The ABL1 kinase domain shows the P-loop, ATP/imatinib binding site, catalytic domain, A-loop, and the most clinically relevant mutations affecting the kinase domain [19]. Figure created in BioRender.com.

The response to TKIs is assessed by measuring the levels of *BCR::ABL1* transcript in the peripheral blood by quantitative real-time PCR (RT-qPCR) and based on the achievement of molecular milestones over time [20]. *BCR::ABL1*% is expressed and reported on a log scale, where 10%, 1%, 0.1%, 0.01%, 0.0032%, and 0.001% corresponds to 1, 2, 3, 4, 4.5, and 5 log reductions, respectively, below the standard baseline used in IRIS study [21]. Both 4-log (MR4) and 4.5-log (MR4.5) reductions are described as deep molecular responses [22]. Optimal molecular response corresponds to achieving specific milestones, which are the early molecular response of *BCR::ABL1* ≤ 10% at 3 months and the major molecular response (MMR) to *BCR::ABL1* ≤ 0.1% at 12 months. The main goal for CML patients is to achieve a durable remission, known as treatment-free remission, which first requires maintaining a deep molecular response (DMR) (*BCR::ABL1* ≤ 0.01% or undetectable; limit of detection of 0.001%) [23]. Patients who achieve a deep molecular response (MR4 or MR4.5) have a better outcome with low risk of disease progression or relapse [22].

Chronic-phase CML patients who achieve an optimal response to treatment can expect a comparable life expectancy to that of the general population. About 25% of patients can cease their TKI therapy and maintain treatment-free remission [24,25,26,27]. Despite this enormous success, several challenges remain. The failure to eradicate persistent CML cells leads to a relapse in about 50% of patients who cease therapy. These patients therefore need to restart therapy, which can cause considerable emotional stress. Additionally, about 20% of CML patients respond poorly to frontline therapy, with 5–10% progressing to blast crisis [28]. This is due to the development of resistance, which represents a “bottleneck to cure” [29]. While a lot of progress has been made in understanding the BCR::ABL1-dependent mechanisms of resistance that rely on BCR::ABL1 reactivation, the mechanisms of BCR::ABL1-independent resistance have remained largely elusive [30]. It is becoming increasingly clear that TKI resistance can be driven by mechanisms that do not depend on BCR::ABL1 activation [31]. These mechanisms need to be considered in the curative approach of CML. In this review, we discuss BCR::ABL1-dependent and BCR::ABL1-independent mechanisms of TKI resistance in CML, highlighting combination-treatment strategies for overcoming resistance in a situation where resistance is driven by a BCR::ABL1-independent mechanism. The combination strategies we explore here could address current treatment challenges and improve treatment outcomes in CML.

## 2. BCR::ABL1-Dependent Mechanisms of Resistance

BCR::ABL1-dependent mechanisms are the most common and well-studied mechanisms of TKI resistance in CML that reactivate the kinase activity of BCR::ABL1 [32,33].

### 2.1. Kinase Domain Mutations

The most common and extensively studied mechanism of secondary resistance to imatinib is due to mutations in the *ABL1* kinase domain [34]. The *ABL1* kinase domain mutations account for approximately 40–60% of CML patients who experience haematological relapse on imatinib therapy [29]. The point mutations (more than 50 different mutations) in the ABL1 kinase domain, including the ATP-binding domain (P-loop), catalytic domain, the activation loop (A-loop), and in amino acids, in the imatinib binding site, are known to be responsible for clinical imatinib resistance (Figure 2C) [35]. The most clinically relevant mutations are G250E, Y253F/H, and E255K/V mutations in the P-loop, T315I in the imatinib binding site, and M351T and F359V/C/I in the catalytic domain (Table 1) [19]. Optimal drug binding requires structural adjustments in BCR::ABL1, which is prevented in P-loop mutants, while the kinase is stabilized in an active state in A-loop mutants [19]. The T315I mutation, also known as “the gatekeeper” mutation, is generated when threonine is replaced with isoleucine, preventing imatinib from forming a hydrogen bond with the protein [19].

The second-generation TKIs, nilotinib and dasatinib, have an increased potency and activity against most imatinib-resistant mutants but are not effective against a T315I mutation (Table 1) [36]. Ponatinib is effective against most kinase domain mutations, including T315I, but there still remains some compound mutations, which are two or more mutations in the same allele (e.g., Y253H/T315I or E255V/T315I) that can confer resistance to ponatinib [37]. The new allosteric inhibitor asciminib was shown to be potent and effective against naive as well as mutated BCR::ABL1 proteins, including the T315I BCR::ABL1 mutation [16,38]. In 2019, Hughes et al. reported the results of the first large phase I study of asciminib as a second line of therapy for CML patients who failed or were intolerant to at least two previous TKIs [16]. Asciminib effectively induced complete haematological responses in 14/16 (88%) and major molecular responses in 4/17 (24%) CML patients with T315I mutations [16]. Similarly, Cortes et al. also reported favourable safety and acceptable clinical efficacy in CML patients with the T315I mutation, showing that around 50% of these patients achieve a major molecular response with asciminib treatment [39]. Moreover, Gutiérrez et al. reported that in CML patients who were heavily pretreated with three or more TKIs prior to asciminib treatment, 48% achieved a complete cytogenic response, and 33% achieved a major molecular response [40]. Pagani et al. reported the results of asciminib treatment in a CML patient with an atypical e19a2 *BCR::ABL1* transcript, who had previously developed a T315I kinase domain mutation. The patient had reached a deep molecular response and maintained it for 4.6 years when the study was published [38]. The first trial of asciminib as a front-line therapy is now currently ongoing and enrolling newly diagnosed CML patients (NCT03578367).

**Table 1 cancers-14-03300-t001:** Clinically important BCR::ABL1 kinase domain mutations and their sensitivity to different TKIs [30,41].

Kinase Domain Mutations	TKI Sensitivity
T315I, Y253F/H, E255K/V, Q252H, M244V, L248V, G250E, F317L, M351T, M355D, F359V, and H396R/P/A	Reduced sensitivity to imatinib
T315I, L248V, Y253H, E255K/V, and F359V/I/C	Reduced sensitivity to nilotinib
T315I/A, V299L, and F317L/V/I/C	Reduced sensitivity to dasatinib
T315I, E255V/K, V299L, G250E, E255K/V, and F317L/V/I/C	Reduced sensitivity to bosutinib
T315M/L	Reduced sensitivity to ponatinib

### 2.2. Myristoyl Domain Mutations

Despite the promising efficacy of asciminib, especially against the T315I mutation, some patients acquire mutations in the myristoyl-binding pocket [23]. They include the A337V, P465S, V468F, I502L, and C464W mutations, which have been shown to confer asciminib resistance while retaining sensitivity to ATP-competitive kinase inhibitors [42,43]. Combining asciminib (50 or 250 nM) lowered the IC50 of ponatinib by 1.9 to 18.5-fold for compound mutants involving the T315I mutation and by 3.1 to 6.3-fold for compound mutants not involving the T315I mutation [43]. Similarly, mutant clones with kinase domain and myristoyl-binding site mutations conferred resistance to asciminib or ponatinib, but resistance was largely overcome when the two treatments were combined [43]. This finding supports the exciting possibility of combining an approved TKI with asciminib for the treatment of patients displaying resistance.

### 2.3. BCR::ABL1 Overexpression

BCR::ABL1 overexpression due to Ph chromosome duplication, *BCR::ABL1* gene amplification, or altered transcription of *BCR::ABL1* gene can occur in some patients. However, the role of BCR::ABL’s overexpression in the context of TKI resistance is not as well understood as the role of kinase domain mutations [44]. One hypothesis is that increased BCR::ABL1 levels may provide cells with sufficient kinase activity for acquiring kinase domain mutations to induce resistance in TKI-treated cells [33]. Therefore, higher BCR::ABL1 expression appears to provide a selective advantage for CML cells. This increased expression is common in advanced-stage CML, where reduced TKI sensitivity and development of resistance is often observed [30].

### 2.4. Altered Expression of Drug Transporters

Another mechanism that potentially prevents complete inhibition of BCR::ABL1 by TKIs is an increase in drug efflux. Intracellular drug concentration is controlled by ATP-binding cassette (ABC) transporters such as P-glycoprotein (ABCB1) and a breast cancer-resistant protein (ABCG2) [45]. Different ABC family transporters were identified to be resistant to different TKIs. For instance, ABCB1 was implicated in imatinib and nilotinib resistance [46,47]; ABCC6 was implicated in nilotinib and dasatinib resistance; and ABCG2 was mainly associated with asciminib resistance [48]. Low activity of organic cation transporter-1 (OCT-1), a cellular influx pump, can reduce the intracellular drug availability, consequently promoting imatinib resistance [49]. However, OCT-1 does not appear to regulate the intracellular uptake of second- and third-generation TKIs [50,51]. The resistance induced by overexpression of a particular transporter could be treated by using a TKI that is not susceptible to that transporter or by adding an inhibitor of that transporter [52]. For instance, Agrawal et al. reported that imatinib-resistant CML patients had high levels of ABCB1 expression but still responded to second-line nilotinib treatment [53]. Similarly, Qiang et al. identified the overexpression of ABCG2 as a major mechanism of resistance in asciminib-resistant K562 cells and showed that an inhibitor of ABCG2, Ko143 (100 nM), was able to restore the effectiveness of asciminib against those cells [43].

## 3. BCR::ABL1-Independent Mechanisms of Resistance/Persistence

Approximately 50% of CML patients with poor responses/disease progression have kinase domain mutations, suggesting that the remaining patients display alternative mechanisms of resistance [54]. This could be due to mechanisms intrinsic to the leukaemic cells that alternatively activate or create de novo parallel bypass survival pathways [55]. Additionally, because TKIs target only differentiated and cycling cells [56], certain leukaemic cells could also “escape” the apoptosis induced by TKIs and become dormant [56]. In this section, we describe the principal BCR::ABL1-independent mechanisms of resistance, including the alternative activation of major BCR::ABL1-driven pathways and other pathways that support alternative survival mechanisms. We also focus on the development of targeted therapies against specific deregulated pathways.

### 3.1. Alternative Activation of MAPK Pathway

The mitogen-activated protein kinase (MAPK) pathway (also known as the RAF/RAS/MEK/ERK pathway or RAS pathway) is a major BCR::ABL1-downstream signalling pathway responsible for cell proliferation, cell survival, and prevention of apoptosis in BCR::ABL1-positive cells (Figure 3) [57]. TKIs can inhibit this pathway by blocking BCR::ABL1 and inducing apoptosis in CML cells. However, the activation of the RAS pathway driven by mechanisms independent of BCR::ABL1 signalling, such as mutations in intermediates of the RAS pathway, can lead to TKI resistance in CML [58,59]. Ma et al. demonstrated that the increased activation of the RAS pathway as a result of *protein kinase c* (*PKC*) overexpression led to BCR::ABL1-independent imatinib resistance in K562 cells, as well as in patient-derived CML leukaemic stem cells [58]. Likewise, Kuman et al. also reported that imatinib-resistant CML leukaemic stem cells in contact with stroma had increased ERK1/2 phosphorylation. The inhibition of ERK1/2 using U0126 (ERK1/2 inhibitor) led to imatinib-induced apoptosis in those cells [60].

### 3.2. Alternative Activation of JAK/STAT Pathway

JAK/STAT signalling is another major BCR::ABL1-downstream signalling pathway that regulates proliferation, survival, and drug resistance in CML (Figure 3) [57,61]. JAK/STAT signalling is mediated by two important effectors, STAT3 and STAT5 [62]. Deininger et al. observed the activation of STAT3 via phosphorylation of Y705, leading to TKI resistance in CML CD34+ cells. A subsequent combination treatment with imatinib and a highly potent and specific STAT3 inhibitor, BP-5-087, reduced the survival of both TKI-resistant leukaemic stem and progenitor cells [63]. Kuepper et al. also observed similar results when BCR::ABL1 and JAK1 were co-inhibited. The JAK-specific inhibitors filgotinib and itacitinib were used to target JAK1 [64]. A significant increase in pSTAT3 Y705 was also identified by using single-cell mass cytometry in nilotinib-treated CML patients. This finding has possible prognostic potential [65]. Extrinsically activated JAK1/STAT3 signalling driven by external stimuli has also been shown to mediate stem cell persistence in CML [64]. Similarly, a high expression of STAT5 has also been shown to significantly reduce the sensitivity of CML cells to TKI-induced apoptosis in in vitro and in vivo studies [66].

### 3.3. Alternative Activation of PI3K/AKT Pathway and Dysregulation of Autophagy

The activation of the PI3K/AKT/mammalian target of rapamycin (mTOR) signalling pathway is another major BCR::ABL1-driven pathway that promotes cell growth and cell proliferation in CML cells (Figure 3) [67]. Burchert et al. showed AKT/mTOR pathway activation as a BCR::ABL1-independent compensatory pathway in imatinib-treated BCR::ABL1-positive LAMA cells [68]. BCR::ABL1-independent PI3K/AKT pathway activation and imatinib resistance was also observed in primary leukaemia cells and in imatinib-treated CML patients [68]. Moreover, Wagle et al. revealed imatinib and dasatinib dual-resistant K562 cells had sustained PI3K/AKT pathways with an elevated forkhead box O1 (FOXO1) level in the cytoplasm. However, the treatment with a class I PI3K inhibitor, GDC-0941 (pictilisib), led to the nuclear translocation of FOXO1 and induction of apoptosis in those cells [69]. Elevated FOXO1 levels were also observed in primary samples from relapsed CML patients who lacked kinase domain mutations [69]. Furthermore, Mitchell et al. showed an alternative activation of mTOR in BCR::ABL1-independent ponatinib-resistant CML cells and demonstrated that the inhibition of mTOR induce autophagy and that the inhibition of autophagy sensitizes those cells to ponatinib treatment [70].

Bellodi et al. demonstrated that BCR::ABL1 suppresses autophagy partly via the PI3K/AKT/mTOR pathway, leading to the downregulation of key autophagy genes *Beclin 1* and *Autophagy related 5* (*ATG5*) [71]. Autophagy is a conserved catabolic process responsible for protein degradation and the mediation of antigen presentation [72]. Autophagy is an important process for the maintenance of normal haematopoietic stem cells, and the dysregulation of autophagy is a common characteristic of leukaemic stem cells [72]. Several studies have described autophagy as a key process in drug resistance, where autophagy helps cells tolerate stress and prevents apoptosis induced by anti-cancer drugs [73]. Michell et al. demonstrated the genetic or pharmacological inhibition of mTOR, and therefore, autophagy primed BCR::ABL1-independent ponatinib-resistant CML cells to the mTOR inhibitor and induced apoptosis [70]. Moreover, Karvela et al. also showed that the knockdown of ATG7, an autophagy protein, sensitized CML progenitor cells to TKI-induced cell death [74]. The survival of normal cells remained unaffected, suggesting this strategy could be used as a novel way to target persistent leukaemic cells in CML [74].

### 3.4. Activation of Wnt/β-Catenin Signalling

The Wnt signalling pathway is necessary for the self-renewal of normal cells, but it has been implicated in cancer progression [55]. The activation of nuclear β-catenin and expression of its transcriptional targets leads to leukaemic progression, TKI resistance, and self-renewal in CML stem cells [55]. The expression of β-catenin is shown to be regulated by BCR::ABL1 via the PI3K/AKT pathway and to enhance the leukaemic progression in a CML murine model (Figure 3) [75]. BCR::ABL1 also led to a reduction in the protein expression of β-catenin antagonist Chibby1, which was more prominent in leukaemic stem cells compared with progenitor cells [76]. However, Eiring et al. demonstrated that leukaemic cells maintained their cytoplasmic β-catenin expression despite BCR::ABL1′s inhibition by imatinib, suggesting that β-catenin expression was decoupled from BCR::ABL1 and contributed to intrinsic TKI resistance in CML cells [55]. In addition, Zhou et al. showed that the overexpression of β-catenin was present in blast crisis CML stem and progenitor cells and that the pharmacological and genetic inhibition of β-catenin impaired the self-renewal of stem and progenitor leukaemic cells [77].

### 3.5. Protein Phosphatase 2A (PP2A) Level

The tumour-suppressor protein phosphatase 2A (PP2A) gene encodes for a multimeric serine/threonine phosphatase and is involved in the regulation of transcription factor β-catenin, apoptosis, and the maintenance of G1/S cyclin levels during cell cycle progression [78,79]. BCR::ABL1 regulates PP2A and inhibits its phosphatase activity via the expression of an endogenous inhibitor, SET (Figure 3) [57,80]. Neviani et al. showed that the re-activation of PP2A was associated with growth arrest and apoptosis in BCR::ABL1-positive cells [80]. Moreover, BCR::ABL1 protein expression was required for JAK2 activation, whereas kinase activity was not; in turn, this activated SET-dependent PP2A inhibition, leading to the expression of β-catenin and subsequent self-renewal and survival signalling in quiescent LSCs [80].

### 3.6. Epigenetic Alterations

Genetic alteration of epigenetic regulator genes such as Additional sex combs-like 1 (ASXL1), DNA (cytosine-5)-methyltransferase 3A (DNMT3A), runt-related transcription factor 1 (RUNX1), and Tet methylcytosine dioxygenase 2 (TET2) are frequently found in a CML blast crisis [81,82]. Somatic mutations in epigenetic regulator genes have been associated with poor TKI response and progression to the advanced stage of disease in CML patients when acquired during TKI therapy [83,84,85]. However, their causal relationship to TKI resistance, disease progression, and relapse is yet to be elucidated. Similarly, expression of the enhancer of zeste homolog 2 (EZH2), an epigenetic re-programmer, has been implicated in TKI resistance in CML. EZH2 is a histone methyltransferase and a catalytic subunit of polycomb repressive complex 2 (PRC2) [86]. In a CML mouse model, Scott et al. demonstrated that EZH2 was overexpressed in LSCs and that its dysregulation was responsible for TKI resistance and LSCs’ protection [87]. The inactivation of EZH2 through CRISPR/Cas9-mediated gene editing led to the reduced initiation, maintenance, and survival of LSCs, irrespective of BCR::ABL1 mutations [86]. The dysregulation of PRC2 has also been demonstrated in CML stem cells, where the re-programming of EZH2 and H3K27me3 led to apoptosis prevention and LSC survival [87].

### 3.7. Inflammatory TNF-α and TGF-β Pathways

Single-cell transcriptomic analysis showed the enrichment of inflammation-related gene expressions such as inflammatory tumour-necrosis factor (TNF)-α and the transforming growth factor (TGF)-β genes in LSCs, compared with normal HSCs [88]. The increase in the pathway activity of TNF-α and TGF-β was associated with increased stem cell quiescence and thereby TKI resistance in poor responder CML patients (patients not achieving MMR) [88]. Giustacchini et al. tested the effect of TNF-α and TGF-β on normal HSCs and CML-LSCs in vitro and observed that TNF-α promoted quiescence in both normal HSCs and CML-LSCs, while TGF-β promoted higher cell division in CML-LSCs [88]. Targeting TNF-α and TGF-β activity could be effective in reducing stem cell quiescence and increasing the possibility of eliminating leukaemic stem cells.

### 3.8. Sonic Hedgehog Pathway Activation

The sonic hedgehog pathway is an evolutionarily conserved signalling pathway that plays a role in embryogenesis, cell proliferation, and growth [89]. It is a master regulator of the self-renewal of normal and leukaemic stem cells [90]. Cancer cells with aberrant hedgehog signalling undergo self-renewal, invasion, proliferation, and survival [89]. Hedgehog ligands act on cancer stem cells, both in a paracrine and an autocrine manner, leading to chemotherapy resistance, cancer cell survival, and relapse [91]. The binding of hedgehog ligands activates smoothened (SMO), which in turn activates the transcription factor GL1 [92]. GL1 modulates the transcription of several target genes in the nucleus, such as genes involved in cell cycle regulation and apoptosis, and promotes the MDM2-dependent degradation of p53 (Figure 3) [90]. Studies demonstrated that the treatment of CML-LSCs with the SMO inhibitor PF-04449913 promoted differentiation and increased their sensitivity to TKI [93,94]. Similarly, Anusha et al. showed that the sonic hedgehog pathway was activated in mutation-independent imatinib-resistant CML-LSCs and that treatment with the inhibitor of this pathway potentiated cells to imatinib treatment [95].

### 3.9. Dysregulation of Apoptotic Proteins’ Expression

The intrinsic apoptotic pathway is regulated by the B-cell lymphoma-2 (Bcl-2) family proteins that maintain mitochondrial membrane integrity and regulate the apoptosis cascade through processes such as apoptosome formation, cytochrome-c release, caspase activation, and cleavage of intracellular targets [96]. The Bcl-2 family proteins consists of pro-apoptotic (e.g., BIM and BAD) and pro-survival/anti-apoptotic (e.g., Bcl-2, Mcl-1, and Bcl-xL) proteins. The balance of pro- and anti-apoptotic proteins determines the activation of Bax and Bak. The activation of Bax and Bak leads to cytochrome-c release and the induction of apoptosis [96]. The dysregulation of anti-apoptotic proteins has been implicated in leukemogenesis, disease progression, and treatment resistance in myeloid leukaemia [96]. The overexpression of anti-apoptotic Bcl-2 protein and reduced expression of BIM have been shown to be involved in TKI resistance in CML [96]. BCR::ABL1 increases the expression of anti-apoptotic proteins such Bcl-2, Bcl-xL, and Mcl-1, which all play a role in preventing apoptosis in leukaemic cells [97]. The dysregulation of anti-apoptotic proteins has been observed in CML [97].

## 4. Targeted Therapies against BCR::ABL1-Independent Resistant Cells

Given the variety of BCR::ABL1-independent mechanisms of resistance, it is now important to focus the studies on the development of the most appropriate targeted therapy for a specific deregulated pathway. This will also involve the development of sensitive techniques for identifying specific biomarkers using phospho-flow and next-generation sequencing at diagnosis or at relapse [98]. Mutations in several genes have been identified in CML patients, such as *ASXL1*, *RUNX1*, *TET2*, *BCL6 Corepressor-Like 1* (*BCORL1*), *GATA-binding factor 2* (*GATA2*), and others [81]. Kim et al. showed a strong correlation between the mutations in *TP53*, *Lysine Methyltransferase 2D* (*KMT2D*), and *TET2* during TKI therapy and treatment failure in CML patients [82]. Additionally, *IKAROS Family Zinc Finger 1* (*IKZF1*) exon deletion has been reported to be associated with poorer outcomes in CML patients [81]. In an interesting study, Chan et al. discovered that mutations converging in one principal driver pathway were able to promote progression toward leukaemia [99]. Conversely, divergent signalling pathways represented a powerful barrier to transformations. Mutations in these divergent pathways prevented leukaemia instead of promoting it [99]. This finding is of principal relevance for the development of new targeted therapies against deregulated pathways in leukaemia. Chan et al. also demonstrated that targeting divergent pathways had a counterproductive effect on cancer progression. It is therefore vital to identify the driver pathway and target it [99]. This supports the possibility of developing drugs that could target different mediators of a driver pathway, increasing the possibility of finding the best precision medicine therapeutic approach.

In the context of BCR::ABL1-independent pathways, combination therapies that target both BCR::ABL1 and alternative survival pathways have the potential to eliminate leukaemic stem cells and sensitize progenitor cells, improving the treatment of CML. For instance, using a CML-mouse model, Shan et al. demonstrated that the inhibition of BCR::ABL1 with TKIs, combined with inhibition of the RAS pathway using trametinib, an FDA-approved MEK inhibitor, synergistically induced cell death in BCR::ABL1-independent MAPK pathway-driven imatinib-resistant CML stem cells [58]. Similarly, the JAK/STAT pathway has also been implicated in the survival of quiescent leukaemic stem cells, and the combination therapy of TKIs and ruxolitinib (JAK2 inhibitor) has been demonstrated to reduce their number in murine xenografts [100]. In a phase I clinical trial (NCT01702064), the combination therapy of nilotinib and ruxolitinib resulted in undetectable *BCR::ABL1* in 40% of CML patients after 6 months, as measured by RT-qPCR [101]. As a result, this combination treatment was recommended for further investigation in a phase II clinical trial [101]. Moreover, Yagi et al. reported that the specific activation of STAT3 in CML and the combined inhibition of JAK1 with tofacitinib and BCR::ABL1 with imatinib synergistically induced anti-tumour effects in CML cells [102]. Likewise, many PI3K/AKT/mTOR pathway inhibitors, including those which have received FDA approval, such as PI3K inhibitors (Idelalisib, Copanlisib, and Duvelisib) and mTOR inhibitors (Everolimus, Sirolimus, and Temsirolimus), have been studied as potential treatments against TKI-resistant CML, and active pre-clinical investigation is ongoing (Figure 3 and Table 2) [67]. A clinical trial is underway for investigating the safety and efficacy of the co-treatment of imatinib and everolimus in CML (NCT00093639).

There are multiple clinical trials assessing the inhibitors of the WNT/β-catenin pathway in various haematological malignancies, including CML [32]. The results from phase I and II clinical trials are currently being evaluated for the safety and efficacy of PRI-724, an inhibitor of WNT/β-catenin pathway, in pancreatic cancer and leukaemia patients [32]. More recently, a combination treatment with dasatinib and okadaic acid (an inhibitor of PP2A) has also been shown to induce cell cycle arrest and apoptosis in the K562 cell line [103]. However, conflicting reports of the pro- and anti-survival role of PP2A warrants further investigation to understand the situations where such a dual role of PP2A exists [104]. Consistent with the previously reported tumour-suppressor role of PP2A, its activating drugs, such as FTY720 and OP449, have been shown to re-sensitize TKI-resistant LSCs to BCR::ABL1 inhibition by targeting the JAK2/PP2A/β-catenin pathway (Figure 3 and Table 2) [105]. Therefore, the combination of TKI and an FDA-approved PP2A activator FTY720 or OP449 has been actively investigated in preclinical studies [105].

Due to the ability of EZH2 to reprogram and prevent apoptosis in CML-leukaemic stem cells, the combination of TKI treatment and an FDA-approved inhibitor of EZH2, Tazemetostat, has also been actively investigated in pre-clinical trials [87]. Similarly, Stobo et al. demonstrated that the first-generation autophagy inhibitor hydroxychloroquine potentiated TKI-induced cell death in CML-leukaemic stem cells, but their recent clinical trial combining hydroxychloroquine with imatinib showed that clinically achievable doses of hydroxychloroquine were not sufficient to accomplish the efficient inhibition of autophagy [106]. Autophagy genes were found to be highly expressed in CML-patient-derived leukaemic stem cells compared with progenitor cells, and the genetic or pharmacological inhibition of autophagy with the second-generation autophagy inhibitor Lys05 resulted in decreased leukaemic cell viability and improved sensitivity to chemotherapy [107]. Therefore, the second-generation FDA-approved autophagy inhibitors Lys05 and PIK-III were combined with TKIs to investigate their effects in preclinical trials (Figure 3) [107].

There are ongoing clinical trials investigating the combination of TKIs and IFN-α to eliminate leukaemic stem cells, since IFN-α can augment immune responses while TKIs inhibit BCR::ABL1 to exert combined effects [108]. Palandri et al. revealed higher complete cytogenetic responses in the imatinib and IFN-α group compared with the imatinib-only group (60% vs. 42%, *p* = 0.003) at 6 months, with a more rapid response rate seen with the combination treatment [85]. However, there was no difference in the complete cytogenetic response rate at 48 months (88% vs. 88%). Another phase II study by the Nordic group showed higher major molecular response rates (82% vs. 54%) with an imatinib and IFN-α combination treatment compared with an imatinib-only treatment [109]. Another clinical trial investigating the combination of bosutinib with interferon is also underway, and results are yet to be published (NCT03831776) [110]. The combination therapy of TKIs and the hedgehog inhibitor trial was discontinued at a very early stage due to toxicity being observed [30]. If the drug could be further refined to minimize toxicity, there may be potential for using this combination treatment in CML.

Carter et al. first discovered that the anti-apoptotic protein Bcl-2 plays a key role in the survival of CML stem cells and progenitor cells, making it an attractive CML target [111]. The combined inhibition of Bcl-2 with venetoclax and TKI provided rationale for the potential curative treatment of CML (Figure 3 and Table 2) [111]. The phase II clinical trial (NCT02689440) combining TKI (dasatinib) and venetoclax in heavily pretreated blast-phase CML showed encouraging results, with a 75% overall response rate [112]. Another phase II clinical trial combining ponatinib, venetoclax, and decitabine in blast-phase CML is also underway, investigating the overall patient response rate (NCT04188405) [112].

**Table 2 cancers-14-03300-t002:** This table shows the BCR::ABL1-independent pathways and their inhibitors that could be used in combination with TKIs, and the stage of their current development for diseases listed that could be repurposed in the treatment of Ph+ leukaemias.

Target Pathway	Inhibitor/s	Stage of Development	Approved/Treated Disease	Ref.
RAS/RAF/MEK/ERK	RAF inhibitor: Vemurafenib and Dabrafenib	FDA approved	BRAF(V600E) melanoma	[113]
	MEK inhibitors: Trametinib and Cobimetinib (in combination with vemurafenib)	FDA approved		
JAK/STAT	JAK1 inhibitor: Rinvoq (upadacitinib) and Cibinqo (abrocitinib)	FDA approved	Myelofibrosis and ovarian cancer	[114,115]
	2. JAK2 inhibitor: Ruxolitinib	FDA approved	2. Refractory moderate to severe atopic dermatitis	
	3. JAK1/2 inhibitor: Baricitinib	FDA approved	3. Rheumatoid arthritis	
PI3K/AKT/mTOR	PI3K delta inhibitor: Idelalisib	FDA approved	Leukaemia and lymphoma	[116,117]
	2. PI3K alpha/delta inhibitor: Copanlisib	FDA approved	2. Relapsed follicular lymphoma	
	3. mTOR inhibitor: Sirolimus	FDA approved	3. Lymphangioleiomyomatosis	
Wnt/β-catenin	CBP/β-catenin antagonist: PRI-724	Phase 2 Clinical Trial (NCT01606579)	Acute myeloid leukaemia and chronic myeloid leukaemia	[118]
Tumour suppressor: PP2A	SET: FTY720 (Fingolimod)	FDA Approved	Multiple myeloma and mantle cell lymphoma	[119,120]
Epigenetic modulator: EZH2	Tazemetostat	FDA Approved	Advanced or metastatic epithelioid sarcoma	[121]
Immune system	IFN-α	FDA Approved	Hairy cell leukaemia, CML, follicular non-Hodgkin lymphoma, melanoma, and AIDS-related Kaposi’s sarcoma	[34]
Hedgehog pathway	Vismodegib (GDC-0449) and Sonidegib (LDE225)	FDA Approved	Basal cell carcinoma and acute myeloid leukaemia	[122]
Intrinsic apoptotic pathway	Venetoclax	FDA Approved	Chronic lymphocytic leukaemia and acute myeloid leukaemia	[123]

## 5. Conclusions

Despite significant improvements in the treatment of CML with the clinical development of TKIs, some patients develop TKI resistance, some progress to blast crisis, and most remain dependent on TKI therapy for long-term disease control. The current strategies for addressing TKI resistance have mainly focused on improving the potency and specificity of the drugs that target BCR::ABL1 and/or on overcoming the resistance driven by mutations in the BCR::ABL1 oncogene. However, this approach may be less effective in many CML patients, who develop resistance despite the effective inhibition of BCR::ABL1 with TKIs, i.e., with BCR::ABL1-independent resistance mechanisms. Novel treatment strategies, such as combining TKIs with other agents that target alternative survival signalling, may be necessary to improve treatment outcomes in those patients. A combination-treatment approach could eradicate leukaemic stem cells to maximize treatment-free remission in CML and improve sensitivity to TKIs in non-responsive CML patients. Therefore, some combination strategies that target CML-leukaemic stem cells could be administered as a front-line treatment, while other CML patients who fail TKIs without evidence of any kinase domain/myristoyl domain mutations should be screened for alternative mechanisms of resistance, such as mutations, that can activate an alternative survival pathway to decide an appropriate combination-treatment approach. These combination approaches could address the current unmet treatment needs in CML by eliminating leukaemic stem cells and sensitizing TKI-resistant progenitor cells, therefore improving treatment outcomes.

## Figures and Tables

**Figure 1 cancers-14-03300-f001:**
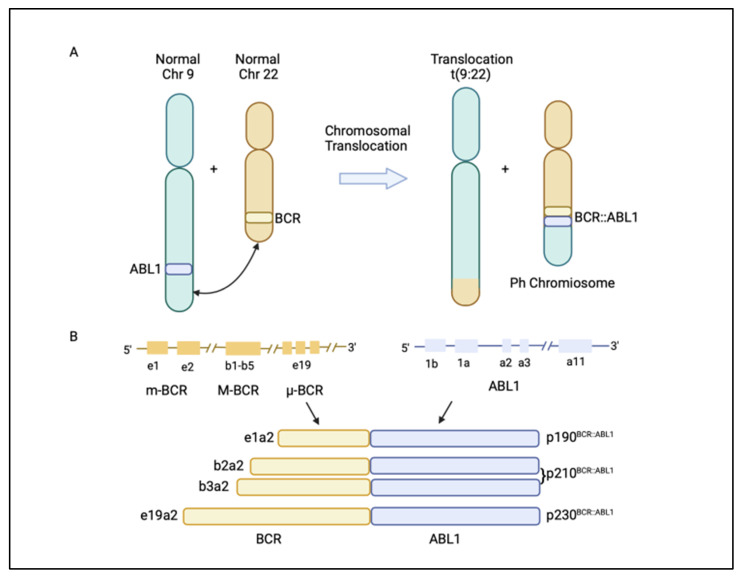
(**A**) Philadelphia (Ph) chromosome is formed from the translocation t(9;22)(q34;q11) of chromosome 9 and 22. The translocation event leads to the fusion of the *breakpoint cluster region* (*BCR*) gene with the *Abelson1 proto-oncogene 1* (*ABL1*) gene, resulting in a *BCR::ABL1* fusion gene. (**B**) Different *BCR::ABL1* fusion gene transcripts p190^BCR::ABL1^, p210^BCR::ABL1^, and p230^BCR::ABL1^ are generated, depending on where the break occurs in the *BCR* gene [5]. Figure created in BioRender.com.

**Figure 3 cancers-14-03300-f003:**
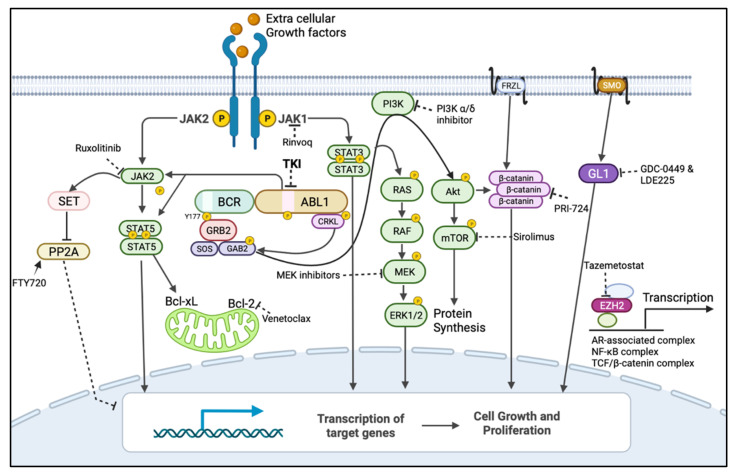
BCR::ABL1-dependent/independent pathways and drugs showing their targets for their BCR::ABL1-independent expression/activation to be used in combination with TKIs. Dark arrow indicates activation, dotted line indicates inhibition by inhibitor/s and encircled p indicates phosphorylation. The mitogen-activated protein kinase (MAPK) pathway (also known as rat sarcoma virus (RAS)/rapidly accelerated fibrosarcoma (RAF)/mitogen-activated kinase kinases (MEK)/extracellular signal-regulated kinase (ERK)), PhosphatidylInositol-3-Kinase (PI3K)/AKT/mammalian target of rapamycin (mTOR) and Janus Tyrosine Kinase (JAK), and signal transducer and activator of transcription (STAT) pathways are the major BCR::ABL1-downstream pathways responsible for BCR::ABL1-independent TKI resistance when re-activated by alternate routes. BCR::ABL1 can mediate the inhibition of the tumour-suppressor protein phosphatase 2A (PP2A) and activation of β-catenin to promote leukaemia, but this can also occur independently from BCR::ABL1. Leukaemic cells can also use JAK1/STAT3, Wnt signalling, the sonic hedgehog pathway, and the expression of the epigenetic modulator EZH2 to remain quiescent, which could contribute to resistance and relapse [30]. These pathways could be targeted by using inhibitors/activators that were developed for other diseases and could be repurposed for a combination treatment with TKIs to treat Ph+ leukaemias. Figure created in BioRender.com.

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
