# Peer review of "Mechanisms of Resistance and Implications for Treatment Strategies in Chronic Myeloid Leukaemia"

_cancers, 2022, doi:10.3390/cancers14143300_

Round 1
Reviewer 1 Report
The Review entitled: " Mechanisms of Resistance and Implications for Treatment Strategies in Chronic Myeloid Leukaemia " is an overview of the most common forms of resistance in BCR/ABL positive cells.
The review has a lot of typos (especially in the conjugation of verbs), sentences not clear, and too long.
Here attached some comments:
· In the introduction it would be appreciated a brief description of the CML pathology, including the clinical features and progression, considering the presence of accelerated phase and not only blast crisis.
· In the introduction, it would be better to include a paragraph (not only in figure 2) describing the molecular activity of BCR/ABL, according to the domains involved. Only like this, it will be easier to understand the mechanisms of action of TKIs and of Myristoyl Domain Mutations.
· In line 85 considering the era of MR4.5 and MR5, a short issue about this concept should be included.
· In paragraph 2, the second sentence is redundant
· In line 255 protein kinase C is PKC and not PKCH.
· In paragraph 3.2 the second sentence is too long and not clear
· Line 255 is not clear. Foxo1 elevation is associated with cytoplasmic retention, so does it mean that it is inactivated?
· Sentence in line 273 is incorrect.
· Titles of paragraphs 3.5 and 3.6 should be changed.
Author Response
Dear Prof Samuel C. Mok
Thank you for the opportunity to submit an amended manuscript. We thank both reviewers for their careful review and for their comments on how we can improve our paper. Our responses to specific comments are below.
We trust that these changes have improved the clarity of the paper.
Yours sincerely,
Govinda Poudel, Molly Tolland, Timothy Hughes, and Ilaria S Pagani
Reviewer 1
Comment 1: The review has a lot of typos (especially in the conjugation of verbs), sentences not clear, and too long.
Response 1: The manuscript has now been thoroughly checked by a native English user. Appropriate changes have been made with special consideration given to the conjugation of verbs, sentence length and clarity.
- Comment 2: In the introduction it would be appreciated a brief description of the CML pathology, including the clinical features and progression, considering the presence of accelerated phase and not only blast crisis.
Response 2: We have now included a brief CML pathology including its clinical features, discussed progression to advance stage disease and considered accelerated phase leading to the blast phase or blast crisis. These changes have been made in the first paragraph of the introduction.
- Comment 3: In the introduction, it would be better to include a paragraph (not only in figure 2) describing the molecular activity of BCR/ABL, according to the domains involved. Only like this, it will be easier to understand the mechanisms of action of TKIs and of Myristoyl Domain Mutations.
Response 3: We have added a paragraph (the second paragraph of the introduction) to describe molecular activity of BCR::ABL1 and explained different domains of BCR and ABL1 involved. This has hopefully made it easier for reader to understand the mechanism of action of TKIs and the role of myristoylated N-terminal in BCR::ABL1 activation.
- Comment 4: In line 85 considering the era of MR4.5 and MR5, a short issue about this concept should be included.
Response 4: We have defined different molecular responses in CML including MR4.5 and MR5 in line 110 to 113 of the amended manuscript. We have also included the significance of achieving those milestones in line 119 and 120 of the updated manuscript.
- Comment 5: In paragraph 2, the second sentence is redundant
Response 5: We have removed second sentence from paragraph 2 and improved first sentence in line 141 and 142 of the updated manuscript.
- Comment 6: In line 255 protein kinase C is PKC and not PKCH.
Response 6: The abbreviation for protein kinase C is now changed to PKC in line 260 of the updated manuscript.
- Comment 7: In paragraph 3.2 the second sentence is too long and not clear
Response 7: We have now divided the second sentence in paragraph 3.2 and removed unnecessary words. Please, refer to highlighted text in paragraph 3.2 in line 270 to 273 of the updated manuscript.
- Comment 8: Line 255 is not clear. Foxo1 elevation is associated with cytoplasmic retention, so does it mean that it is inactivated?
Response 8: We have made the statement in line 255 (now line 289 to 294 of the updated manuscript) clearer by explaining that activation of PI3K/AKT pathway can lead to FOXO1 retention in cytoplasm and TKI resistance. Inhibition of PI3K/AKT pathway can induce translocation of FOXO1 to the nucleus and trigger apoptosis in those cells.
- Comment 9: Sentence in line 273 is incorrect
Response 9: We have now corrected sentence in line 273. Please, refer to highlighted text in line 306 to 309 of the updated manuscript.
- Comment 10: Titles of paragraphs 3.5 and 3.6 should be changed.
Response 10: We have now changed the title of paragraph 3.5 from ‘Inhibition of Protein Phosphatase 2A (PP2A)’ to ‘Protein Phosphatase 2A (PP2A) level’ and paragraph 3.6 from ‘EZH2 Expression’ to ‘Epigenetic Alterations’.
Reviewer 2 Report
There is now ample evidence that epigenetic dysregulation contributes to leukemic stem cell generation, maintenance, and progression in CML. Since you have mentioned several epigenetic regulating genes in line 361-364, I recommend that the “3.6 EZH2 Expression” could change to “Epigenetic Alterations” and add some epigenetic regulating genes like DNMT3A, TET2, ASXL1, etc.
These references below could be taken into consderation.
Kim, T.; Tyndel, M.S.; Kim, H.J.; Ahn, J.-S.; Choi, S.H.; Park, H.J.; Kim, Y.-K.; Kim, S.Y.; Lipton, J.H.; Zhang, Z.; et al. Spectrum of somatic mutation dynamics in chronic myeloid leukemia following tyrosine kinase inhibitor therapy. Blood 2017, 129, 38–47.
Mitani, K.; Nagata, Y.; Sasaki, K.; Yoshida, K.; Chiba, K.; Tanaka, H.; Shiraishi, Y.; Miyano, S.; Makishima, H.; Nakamura, Y.; et al. Somatic mosaicism in chronic myeloid leukemia in remission. Blood 2016, 128, 2863–2866.
Togasaki, E.; Takeda, J.; Yoshida, K.; Shiozawa, Y.; Takeuchi, M.; Oshima, M.; Saraya, A.; Iwama, A.; Yokote, K.; Sakaida, E.; et al. Frequent somatic mutations in epigenetic regulators in newly diagnosed chronic myeloid leukemia. Blood Cancer J. 2017, 7, e559
Author Response
Dear Prof Samuel C. Mok
Thank you for the opportunity to submit an amended manuscript. We thank both reviewers for their careful review and for their comments on how we can improve our paper. Our responses to specific comments are below.
We trust that these changes have improved the clarity of the paper.
Yours sincerely,
Govinda Poudel, Molly Tolland, Timothy Hughes, and Ilaria S Pagani
Reviewer 2
Comment: There is now ample evidence that epigenetic dysregulation contributes to leukemic stem cell generation, maintenance, and progression in CML. Since you have mentioned several epigenetic regulating genes in line 361-364, I recommend that the “3.6 EZH2 Expression” could change to “Epigenetic Alterations” and add some epigenetic regulating genes like DNMT3A, TET2, ASXL1, etc.
Response: We have changed the title of paragraph 3.6 to “Epigenetic Alterations” and included some of the epigenetic genes in the text. Please, refer to the line 340 to 346 of the updated manuscript.
Round 2
Reviewer 1 Report
The review now is better, scientifically sound, and better organized.